# High precision FRET studies reveal reversible transitions in nucleosomes between microseconds and minutes

Alexander Gansen[1], Suren Felekyan[2], Ralf Kühnemuth[2], Kathrin Lehmann [1,2], Katalin Tóth[1], Claus A. M. Seidel [2] & Jörg Langowski [1]

Nucleosomes play a dual role in compacting the genome and regulating the access to DNA. To unravel the underlying mechanism, we study fluorescently labeled mononucleosomes by multi-parameter FRET measurements and characterize their structural and dynamic heterogeneity upon NaCl-induced destabilization. Species-selective fluorescence lifetime analysis and dynamic photon distribution analysis reveal intermediates during nucleosome opening and lead to a coherent structural and kinetic model. In dynamic octasomes and hexasomes the interface between the H2A-H2B dimers and the $(H3-H4)_2$ tetramer opens asymmetrically by an angle of $\approx 20°$ on a 50 and 15 $\mu s$ time scale, respectively. This is followed by a slower stepwise release of the dimers coupled with DNA unwrapping. A mutation (H2A-R81A) at the interface between H2A and H3 facilitates initial opening, confirming the central role of the dimer:tetramer interface for nucleosome stability. Partially opened states such as those described here might serve as convenient nucleation sites for DNA-recognizing proteins.

[1] DKFZ, Div. Biophysics of Macromolecules, 69120, Heidelberg, Germany. [2] Heinrich-Heine-Universität, Lehrstuhl für Molekulare Physikalische Chemie, Düsseldorf 40225, Germany. These authors contributed equally: Alexander Gansen, Suren Felekyan, Ralf Kühnemuth, Kathrin Lehmann. Correspondence and requests for materials should be addressed to K.Tót. (email: kt@Dkfz-Heidelberg.de) or to C.A.M.S. (email: cseidel@hhu.de)

The eukaryotic genome is packaged into nucleosomes, in which about 150 base pairs of DNA are wrapped around a histone protein core[1,2]. These are connected by short stretches (10–90 bp) of linker DNA to form a bead-on-a-string structure, which condenses into higher-order chromatin with the help of linker histones.

The nucleosome provides controlled access to DNA while keeping the compactness and integrity of the genetic material. Transient opening through thermal fluctuations can expose DNA binding sites[3–5] even inside the nucleosome, where DNA unwrapping is much slower[6]. Other mechanisms, e.g. nucleosome repositioning or DNA replication[7] may partially disrupt the nucleosome and help accessing buried DNA. The structure and dynamics of such transient intermediates are still poorly understood.

Single-molecule (sm) techniques allow one to perform a detailed structural and kinetic analysis of transient structures in an ensemble under equilibrium conditions. Single-molecule Förster resonance energy transfer (smFRET) has been applied to studies of nucleosomes[8–18] and nucleosome arrays with respect to structure and dynamics[19,20]. Combined with multi-parameter fluorescence detection (MFD), which classifies the fluorescence signal according to spectral response, lifetime and polarization, smFRET permits deep insight into the structural dynamics of biomolecules[21–26], enabling detection of conformational substates during nucleosome opening[12]. Analyzing nucleosomes with different donor/acceptor positions by smFRET during salt-induced dissociation previously led us to postulate an intermediate open state with all histones still attached[14,17], populated to about 1% under physiological conditions. Similar open structures were later confirmed by others[27–29].

Recently a non-equilibrium stopped-flow ensemble study with small-angle X-ray scattering and ensemble FRET[30] analysis (time resolution >5 ms) examined the kinetics of nucleosome disassembly by diluting to very high salt concentrations (between 1.2 and 1.9 M NaCl). The authors found evidence for unwrapped states with transition kinetics in the range of 50 milliseconds to seconds and suggested an ensemble of different extended DNA conformations formed by asymmetric and symmetric unwrapping with rather extended linker DNA arms where at least half the DNA was unwrapped from the histone core (see Fig. 4 in ref. [30]).

One could argue that the conditions, up to >1 M NaCl, are often outside the physiological range, but the detected intermediates may also be relevant for in vivo dissociation pathways. Here we use a direct approach with a network of four distinct FRET pairs and multi-parameter smFRET[31] to unravel the very first opening steps of mononucleosomes reconstituted on the Widom 601 positioning sequence ("601 nucleosomes")[32] under equilibrium conditions on very short time scales in the microsecond range. If the sample solution is stable and the measurement time is long enough, it has been shown by many groups that single-molecule techniques are perfectly suited to study the kinetics of exchange reactions in biomolecular systems[33].

Based on the detailed insight into structural fluctuations, we derive a comprehensive kinetic scheme for nucleosome disassembly by systematically varying nucleosome concentration, ionic strength, label position and via introducing an H2A point mutation. We conclude that disassembly starts with breathing DNA and formation of dynamic "butterfly" octasomes[14] opening on a 50 μs time scale by an angle of ≈20° each that proceeds through weakening or disrupting of one or both interfaces between the H2A-H2B dimers and the (H3-H4)$_2$ tetramer. Eviction of one H2A-H2B dimer leads to a dynamic hexasome with opening dynamics on a 15 μs time scale which is followed by stepwise asymmetric DNA unwrapping and a slower release of the second dimer. While the loss of the first dimer occurs fast (<300 s at 400 mM), the release of the second dimer is slow (>10,000 s). Our findings provide insight into the mechanism of nucleosome opening, leading to a kinetic scheme for nucleosome disassembly through at least seven species from initial fast unwrapping on the microsecond time scale up to the final dissociation during seconds or minutes.

## Results

**Nucleosome constructs.** To analyze the dynamics of nucleosomes reconstituted on the non-palindromic Widom 601 sequence, we designed four differently labeled nucleosome constructs and one H2A mutant to study dimer release, asymmetric DNA opening and transient intermediates (Fig. 1 and Methods). To account for the asymmetry of the constructs, we denote the left side of the DNA sequence (forward (−) strand) by α and the other side ((+) strand) by β. Analogously, we refer to the (H2A-H2B) dimer bound at the α-side and β-side as Dimer$_\alpha$ and Dimer$_\beta$, respectively.

**Disassembly via asymmetric dimer loss and hexasome formation.** Since nucleosome disassembly under force seems to be initiated by terminal DNA unwrapping followed by H2A-H2B dimer eviction[34], we used the nucleosome construct H2B-Dy$_\alpha$ (Figs. 1, 2a) to analyze nucleosome disassembly with respect to the DNA ends. We followed salt-induced dimer loss measured by FRET between a donor on each H2A-H2B dimer and an acceptor on a DNA site near the dyad axis. We labeled the H2B with an efficiency of ≈45%, so that we could study nucleosomes with one or two donors together with one or no acceptor. To compute the expected distances between the flexibly linked labels and the average FRET efficiency, we modeled dye position distributions (green and red clouds in Fig. 2a) by accessible volume (AV) simulations from the FRET-restrained positioning and screening (FPS) toolkit[23,26]. As the observed donor-acceptor (DA) distances are usually spatially averaged, the interdye distances recovered by fluorescence lifetime- or intensity-based FRET measurements differ slightly due to inherent different weighting. However, the knowledge of the AVs allows us to interconvert them (Methods and Supplementary Table 1). For convenience we present only the FRET-averaged DA distances, $\langle R_{DA}\rangle_E$. In the intact nucleosome $\langle R_{D_\alpha A}\rangle_E$ for the labeled dimer at the α-side is ≈60 Å while $\langle R_{D_\beta A}\rangle_E$ of the labeled dimer at the β-side is ≈79 Å (Fig. 2a) and we could distinguish both dimers in the nucleosomes (Fig. 2b).

We used pulsed interleaved excitation (PIE) in a confocal microscope with multi-parameter fluorescence detection (MFD)[22] to analyze single diffusing nucleosomes (concentration 20 pM labeled in presence of 980 pM unlabeled nucleosomes) by their FRET properties and donor-to-acceptor dye stoichiometry $S$[35] values at different NaCl concentrations. In Fig. 2b we show a representative two-dimensional burst frequency histogram of FRET efficiency, $E$, versus stoichiometry, $S$, for 150 mM NaCl. Nucleosomes with a single donor, D$_\alpha$A or D$_\beta$A, and with two donors D$_\alpha$D$_\beta$A were well resolved. With increasing NaCl concentration all three FRET-active species diminish in favor of the donor-only species (Fig. 2c). The double-donor D$_\alpha$D$_\beta$A subspecies consists of intact nucleosomes which transit from the octasome into the hexasome by loss of one H2A-H2B dimer with $c_{1/2} = 523 \pm 37$ mM NaCl, $c_{1/2}$ being the NaCl concentration at 50% dissociation as measure for nucleosome stability. The single-donor subspecies showed different stabilities against salt: $c_{1/2}$ (D$_\alpha$A) = 546 ± 39 mM and $c_{1/2}$ (D$_\beta$A) = 866 ± 33 mM, indicating that the H2A–H2B dimer preferably evicts from the α-side. This agrees with the asymmetric dissociation of the DNA (Fig. 3). There we compared salt-dependent DNA unwrapping of the two

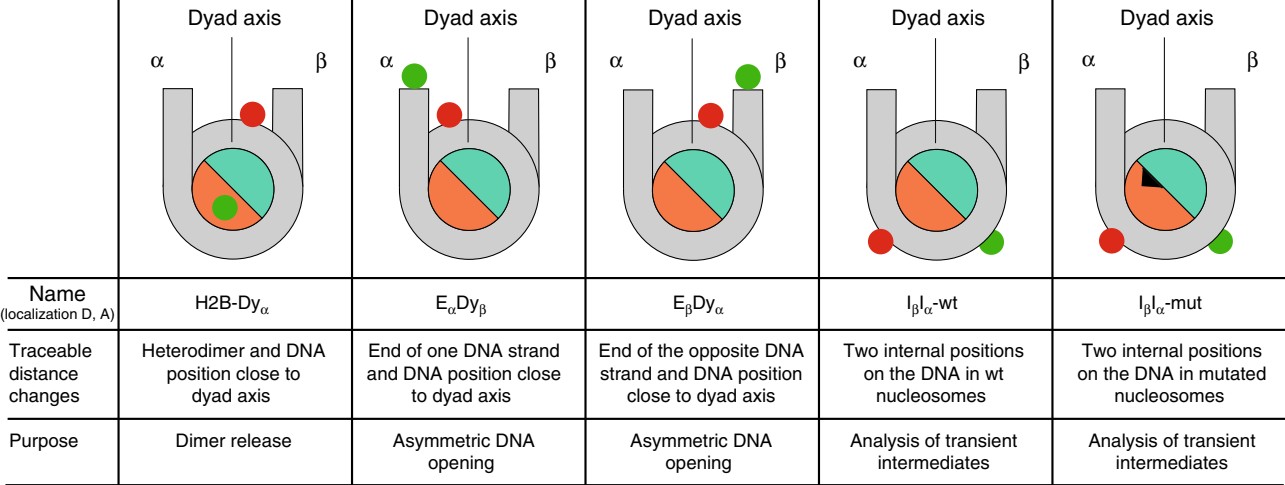

**Fig. 1** Overview on FRET dye positions in the nucleosome constructs. The names of the constructs assign the position and side of the fluorescent labels: first the donor (green circle) and second the acceptor (red circle). In the cartoon representations, the H2A-H2B dimers are shown in orange, the (H3-H4)$_2$ tetramer in turquoise, DNA in gray. To account for the asymmetry, we denote by $\alpha$ the left side of the DNA sequence (forward (-) strand) with base pairs counted in negative numbers from the fragment center (position 0). The other side is called $\beta$, with base pairs counted in positive numbers ((+) strand). The dye labeling positions on the DNA are given as relative base shifts to the middle of the sequence: Dy acceptor close to dyad axis at positions $-15$ and $+15$, E donor labeling position at the end of the DNA strand at positions $-85$ and $+85$, I internal labeling position on the DNA $+41$ (donor) and $-53$ (acceptor), H2B donor at position 112 of H2B using the H2B-T112C mutant, mut the H2A-R81A mutation is represented by a black triangle on the heterodimer

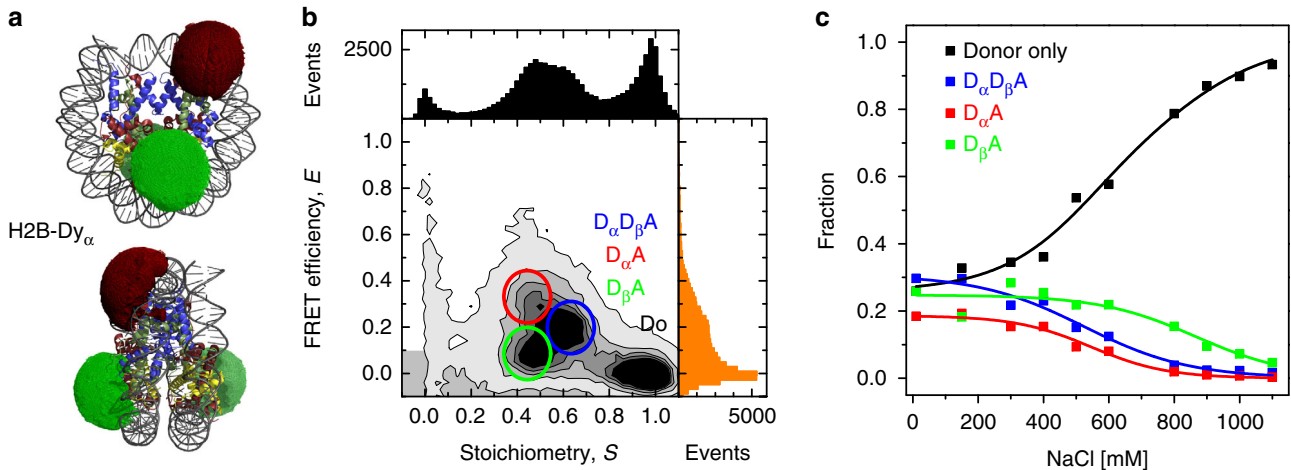

**Fig. 2** MFD-PIE analysis of H2A-H2B dimer release. The nucleosome construct H2B-Dy$_\alpha$ labeled with Alexa488 on each H2A-H2B dimer and Cy5 on the DNA was used. **a** Crystal structure (3LZ1) with accessible fluorophore volumes for H2B-Dy$_\alpha$ nucleosomes (green and red clouds for Alexa488 and Cy5, respectively) that describe the spatial distributions of the dyes. Histones are shown in blue (H3), pale green (H4), ruby (H2A), and yellow (H2B). **b** FRET efficiency versus stoichiometry, $S$, for 20 pM H2B-Dy$_\alpha$ and 980 pM unlabeled nucleosomes at 150 mM NaCl. Subspecies are identified according to their position in the 2D distribution. **c** Relative population of donor-only, D$_\alpha$A, D$_\beta$A, and D$_\alpha$D$_\beta$A for 20 pM H2B-Dy$_\alpha$ and 980 pM unlabeled nucleosomes as a function of NaCl concentration with fit to a sigmoid function (Supplementary Note 5, Supplementary Equation 36, global fit with $x_{\text{Donor only}} = 1 - x_{\text{D}\alpha\text{D}\beta\text{A}} - x_{\text{D}\alpha\text{A}} - x_{\text{D}\beta\text{A}}$). D$_\beta$A nucleosomes prevail at significantly higher NaCl concentration than D$_\alpha$A nucleosomes, indicating that H2A-H2B heterodimers are predominantly evicted from the $\alpha$-side (see details in text)

nucleosome ends by multi-plate ensemble FRET measurements[36] on constructs with double labeled DNA, E$_\alpha$Dy$_\beta$ and E$_\beta$Dy$_\alpha$ (Supplementary Methods). Labeled nucleosomes were diluted to 600 pM and incubated at different NaCl concentrations for 60 min. Both constructs behave similarly in the low-NaCl region (<400 mM), where reversible DNA unwrapping at both ends is expected. We observe differences, though, at higher NaCl concentration where the loss of FRET indicates permanent detachment of the DNA ends from the histone core, likely accompanied by H2A–H2B dimer eviction. The $\alpha$-side was less stably bound ($c_{1/2}$ (E$_\alpha$Dy$_\beta$) = 631 ± 6 mM) than the $\beta$-side ($c_{1/2}$

(E$_\beta$Dy$_\alpha$) = 694 ± 12 mM). Thus, the $c_{1/2}$ values indicate that hexasome formation proceeds preferentially by dimer eviction from the $\alpha$-side. Note that the $c_{1/2}$ values derived from species selective PIE-MFD and from ensemble measurements differ slightly because we probed different regions and species of the nucleosome (see below and Supplementary Note 5).

**Dynamic intermediates are formed with altered DNA geometry.** To analyze the kinetics of transient intermediates during disassembly we used internally labeled I$_\beta$I$_\alpha$-wt nucleosomes with

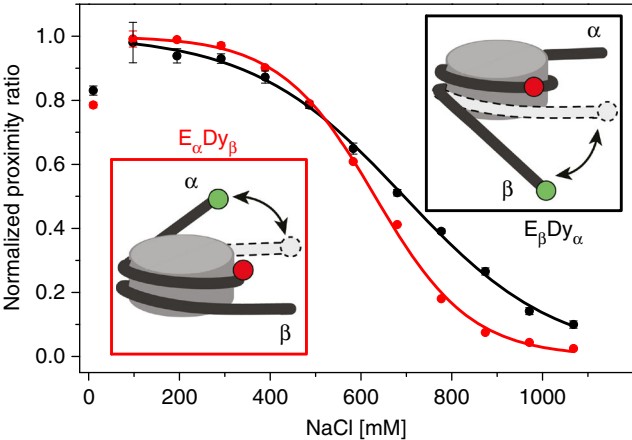

**Fig. 3** Asymmetric unwrapping of DNA ends. Results of DNA unwrapping in the nucleosome constructs $E_\alpha Dy_\beta$ and $E_\beta Dy_\alpha$ (see Methods and Supplementary Note 5) studied by ensemble FRET studies. The dyes were placed at the DNA ends and near the dyad axis to report on unwrapping of each side through loss of FRET. For visualization, data were normalized to the maximal value of the proximity ratio at 100 mM NaCl and approximated by a sigmoidal function (Supplementary Note 5, Supplementary Equation 36) to obtain the $c_{1/2}$ salt concentration. The β-side was significantly more resilient against salt-induced unwrapping than the α-side. Error bars are standard errors from three replicates

both dyes attached to the dimer-binding region of DNA (Fig. 4a). For single-molecule MFD we first diluted the nucleosomes to a concentration of 20 pM at 150 mM NaCl where intermediate structures appear at a physiological salt concentration[12,17]. Figure 4b shows representative burst frequency populations for the burstwise analysis of intensity-derived FRET efficiency, $E$, and average fluorescence-weighted donor lifetime, $\langle\tau_D\rangle_F$, that allowed us to detect the presence of a fast dynamic exchange during the dwell time of a single particle in the detection volume[37]. The correlation between $E$ and $\langle\tau_D\rangle_F$ for molecules that remain in a single static conformation during detection is described by the black static FRET line (Supplementary Methods, Supplementary Equation 40). Any population that is shifted from this line to the right indicates that molecules fluctuate between multiple FRET species during the burst. The data reveal two quasi-static populations with the FRET species, LF (low FRET) and MF (mid FRET), that fall on the static FRET line with mean donor lifetimes of $\langle\tau_D\rangle_F^{LF} \approx 4.0$ ns and $\langle\tau_D\rangle_F^{MF} \approx 2.9$ ns, respectively. A third population is slightly, but significantly, displaced from the static FRET line so that we refer to this broadened peak as the dynF population.

To resolve the composition of these populations with nanosecond time resolution, we accumulated subsets of selected single-molecule events belonging to a single population in Fig. 4b (highlighted in Supplementary Fig. 1a) and performed a global sub-ensemble analysis of their donor fluorescence decays (Supplementary Note 1, Supplementary Figs.1c-e). In agreement with the FRET-lines, we detected three global FRET species by their characteristic donor lifetimes and corresponding DA distances, $\langle R_{DA}\rangle_E$: LF with $\langle\tau_D\rangle_F^{LF} \approx 4.00$ ns ($\langle R_{DA}\rangle_E \approx 97$ Å), MF* with $\langle\tau_D\rangle_F^{MF*} \approx 2.65$ ns ($\langle R_{DA}\rangle_E \approx 57$ Å) and HF with $\langle\tau_D\rangle_F^{HF} \approx 1.30$ ns ($\langle R_{DA}\rangle_E \approx 46$ Å) (Supplementary Equation 1, Supplementary Figs. 1c-e, Supplementary Tables 2-3). Considering the two quasi-static populations LF and MF, the corresponding FRET-species fractions recovered for LF (93%) and MF (79%), respectively, indicate that these populations are dominated by a single FRET species. The dynF population, however, consisted of a mixture of the FRET-species: HF and MF* that

are in fast dynamic exchange. The exchange is well described by the dynamic FRET line connecting HF and MF* (violet curve in Fig. 4b, Supplementary Methods, Supplementary Equation 41). The existence of this second transiently populated mid-FRET species (MF*) which is in slow exchange with MF (yellow curve in Fig. 4b) is further justified at the end of this section. On the time scale of diffusion (3 ms) we did not observe dynamic mixing between MF and LF, suggesting that MF ⇄ LF transitions are significantly slower. Thus, LF can be considered static.

To extract absolute interdye distances and transition rate constants on the sub-millisecond time scale from the measured smFRET distributions, we characterized these subspecies further by dynamic photon distribution analysis (dynPDA)[37] of the D and A fluorescence intensities[38]. DynPDA utilizes the stochastic nature of photon emission, background signal, spectral crosstalk, and dynamic exchange between states. We modeled the system with two exchanging states, MF* and HF, two static populations, MF and LF, and donor-only events (Fig. 4d). Since fluorescence lifetimes of MF and MF* were similar, we approximated the DA distances of the MF and MF* species by a joint Gaussian distance distribution (MF, MF*) with the FRET-averaged distance $\langle R_{DA}^{MF}\rangle_E$ and a half width $\sigma = 2$ Å (Supplementary Fig. 2e). The average interdye distances from 4 independent measurements were $\langle R_{DA}^{LF}\rangle_E = (85.8 \pm 3.2)$ Å, $\langle R_{DA}^{MF}\rangle_E = (61.1 \pm 3.0)$ Å and $\langle R_{DA}^{HF}\rangle_E = (46.1 \pm 0.8)$ Å, which agrees well with the value obtained by sub-ensemble fluorescence lifetime analysis of the dynF population (Supplementary Tables 2 and 3). At the concentration of 20 pM nucleosome and 150 mM NaCl, the forward and backward rates of the MF* ⇄ HF transition were $k_{MF*\rightarrow HF} = (33.4 \pm 3.2)$ ms$^{-1}$ and $k_{HF\rightarrow MF*} = (13.2 \pm 2.8)$ ms$^{-1}$ with a corresponding relaxation time $t_R = (21.6 \pm 1.4)$ μs.

In summary, we assign the dominant species MF to the intact nucleosome (see Methods), since $\langle R_{DA}^{MF}\rangle_E \approx 61$ Å from dynPDA agrees very well with the distance estimated from the crystal structure 3LZ1 using the FPS toolkit ($\langle R_{DA}\rangle_E = 60.3$ Å). LF is attributed to an open conformation, where the DNA is almost completely unbound. HF is considered as an intermediate species during disassembly with a lifetime in the range of tens of microseconds.

We have several corroborating arguments for the coexistence of MF and MF*: (i) Within the precision of FRET the 2D MFD data of donor lifetime and intensity-based FRET efficiency indicate that the interdye distance of MF* is slightly shorter (by ≈4 Å) compared to MF (Fig. 4b). Because the distances are close to $R_0$, this corresponds to a significant change of FRET efficiency $\Delta E \approx 0.1$ that is readily detectable by sub-ensemble fluorescence decay analysis (Supplementary Figs. 1c-e). (ii) The dependence of the kinetic exchange rates on the overall nucleosome concentration, as described next, can only be explained by at least two species with very similar FRET properties: a dynamic (MF*) and a static (MF) subpopulation. (iii) Considering the disassembly pathway of mononucleosomes, mechanistic arguments postulate the existence of two structural states, octasome and hexasome, that have similar FRET efficiencies for the $I_\beta I_\alpha$ FRET-pair (see discussion).

**Dissociation/association of H2A–H2B dimers in intermediates.** To reveal the nucleosome species involved in the MF*-HF transition, we investigated the dependence of this dynamic heterogeneity at 150 mM NaCl by varying the nucleosome concentration with unlabeled sample. DynPDA was employed to resolve the fractions and transition rate constants between the species. Typical MFD data at a total nucleosome concentration of 2 nM are shown in Fig. 4c. The same three subspecies (LF, MF, dynF) were observed as at 20 pM nucleosome concentration but with different

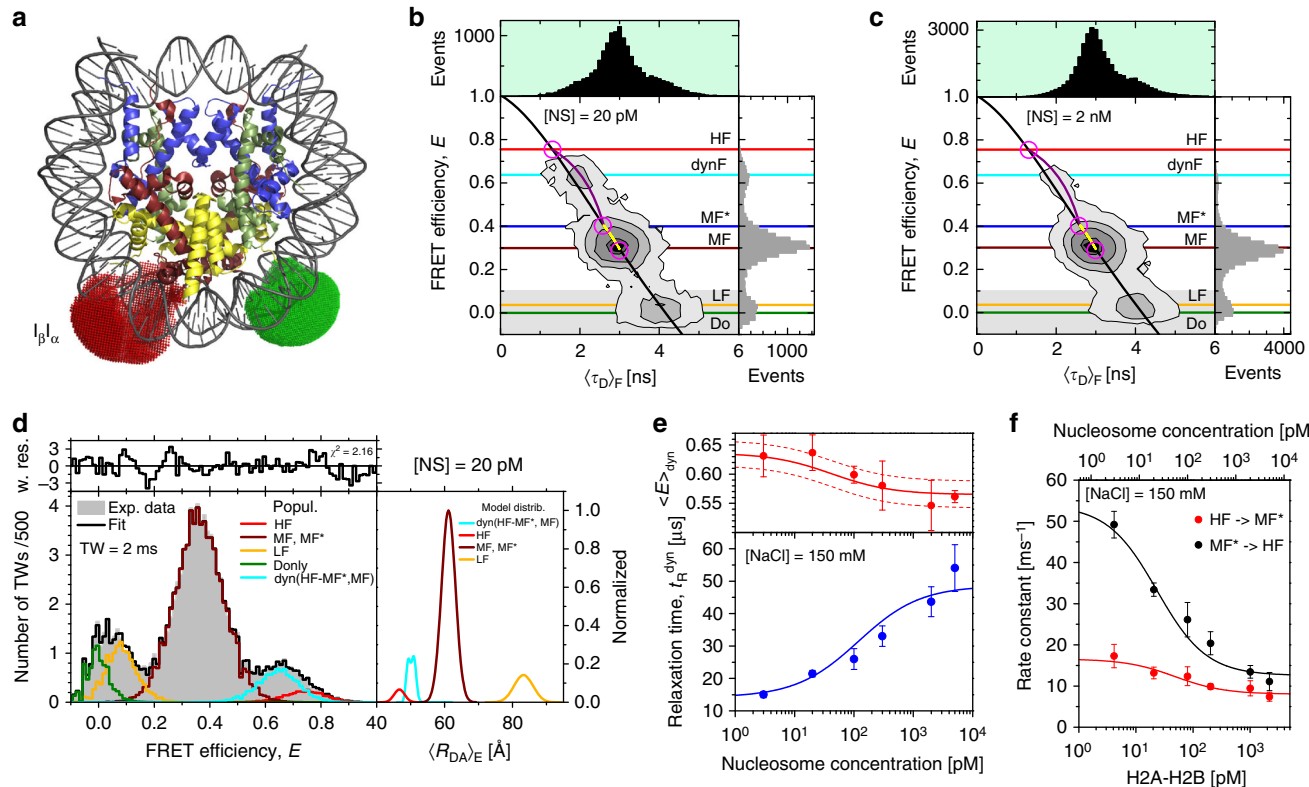

**Fig. 4** Analysis of MFD smFRET data. Results of $I_\beta I_\alpha$-wt nucleosome construct studied at 150 mM NaCl labeled with Alexa488 (donor) and Alexa594 (acceptor). **a** Crystal structure 3LZ1 and accessible fluorophore volumes for donor (green) and acceptor (red). Histones are shown in blue (H3), pale green (H4), ruby (H2A) and yellow (H2B). **b**, **c** FRET efficiency ($E$) versus donor lifetime ($\langle\tau_D\rangle_F$) for 20 pM and 2 nM (20 pM labeled and 1.98 nM unlabeled) nucleosomes. Color coded horizontal lines correspond to FRET efficiency levels of different species. The $E$ - $\langle\tau_D\rangle_F$ relations are presented by black line for static FRET species (Supplementary Methods, Supplementary Equation 40), violet line - for dynamic interconversion between HF ($\langle\tau_D\rangle_F$(HF) = 1.3 ns) and MF* ($\langle\tau_D\rangle_F$(MF*) = 2.65 ns) states and by yellow hypothetical dynamic FRET line connecting MF* and MF. **d** dynPDA of data shown in **b**. Experimental FRET efficiency histogram for 2 ms TW and fitted distributions for a model with two interconverting species (MF* and HF) and three static components as Gaussian-distributed distances (MF, LF, and donor-only). The quality of the fit is demonstrated by weighted residuals in the upper panel. The right panel shows the model with the intrinsic distribution of $\langle R_{DA}\rangle_E$ for the limit of large photon counts (shot noise free). **e** Relaxation times $t_R^{dyn} = (k_{MF^*\to HF} + k_{HF\to MF^*})^{-1}$ and mean FRET efficiency $\langle E\rangle_{dyn} = x_{MF}E^{MF} + x_{HF}E^{HF}$ of dynamic species as a function of nucleosome concentration, calculated from dynPDA fits. $x_{MF(HF)}$ were obtained from rate constants in **f** whereas $E^{MF(HF)}$ were calculated from mean distances of interconverting states ($\langle R_{DA}^{MF}\rangle_E = 61.2 \pm 1.5$ Å and $\langle R_{DA}^{HF}\rangle_E = 46.7 \pm 0.8$ Å). **f** Kinetic rate constants for the MF*⇌HF transition as a function of the free H2A-H2B concentration. Free H2A-H2B is estimated from the total nucleosome concentration and the different species fractions as obtained by PDA. The decrease of $k_{MF^*\to HF}$ with increasing free heterodimer concentration implies the existence of two different MF* species. Error bars are standard errors from at least three different measurements. Lines are a weighted global fit of a four-state kinetic model to the data (see Supplementary Equations 16, 17)

proportions. At higher concentrations intact (MF) nucleosomes are stabilized at the expense of open (LF) and dynamic states (dynF). Interdye distances $\langle R_{DA}^{MF}\rangle_E$ and $\langle R_{DA}^{LF}\rangle_E$ were independent of sample concentration, indicating that the MF and LF structures remained unchanged (Supplementary Fig. 2a). Importantly, the average FRET efficiency $\langle E\rangle$ for the dynF peak decreased by 0.08 with increasing nucleosome concentration (Fig. 4e, top), while the relaxation time for the MF* ⇌ HF transition increased approximately by a factor of 3 (Fig. 4e, bottom).

This concentration dependence implies that the transition from MF* to HF involves an association/dissociation equilibrium, most likely the uptake/eviction of an H2A-H2B dimer. Even if this process were diffusion limited, it would have relaxation times of ms or slower at sub-nanomolar concentrations. Thus, a single MF* ⇌ HF transition is not compatible with the observed fast microsecond kinetics. The simplest reaction scheme that accounts for all observations needs two dynF sub-populations (i.e. four nucleosome states). Each is characterized by a separate MF*-state in fast equilibrium with its HF counterpart: (i) an octasome, O, that contains all core histones, with the equilibrium $O_{cl}^{MF^*} \rightleftharpoons O_{op}^{HF}$

and (ii) a hexasome, H, that misses one H2A-H2B dimer, with the equilibrium $H_{cl}^{MF^*} \rightleftharpoons H_{op}^{HF}$ (two dashed vertical boxes in the kinetic scheme later on). Dynamic states have the indices cl for the closed and op for the open conformation. Additionally, $O_{st}^{MF}$ corresponds to the static octasome (PDB ID: 3LZ1) that is the majorly populated mid-FRET species MF. The MF* species does not appear as a static subpopulation in the 2D MFD analysis, implying that they are in fast equilibrium with their HF counterparts and are not observable separately, but form a part of the dynF population. Based on the characteristic concentration dependence of the FRET efficiencies and anisotropy values (Supplementary Table 4) and recalling that the preferential dimer loss occurred at the α-side (Fig. 2c), we hypothesize that the same FRET species, MF* ⇌ HF, monitor fast parallel transitions within the dynamic octasome and the hexasome, respectively. The slow eviction/uptake of one H2A-H2B dimer that can occur at sub-nanomolar concentrations[14] couples these two equilibria.

The states, $O_{cl}^{MF^*}$, $H_{cl}^{MF^*}$ and $O_{st}^{MF}$, have very similar FRET efficiency values and are therefore difficult to distinguish by static FRET studies alone. However, the global analysis by dynPDA

using this four state model (Supplementary Note 2) and the sub-ensemble lifetime data suggest a simple reason for the concentration dependence of $k_{MF^*\to HF}$ in Fig. 4f: the forward rate $k_{MF^*\to HF}$ in the dynamic octasome population is slower than in the hexasome population. Since higher nucleosome concentrations decrease the hexasome population by pushing the majority of nucleosomes back via the dynamic octasome population into the intact, closed octasome species $(O_{st}^{MF})$, $k_{MF^*\to HF}$ drops. In contrast, the rate constant for the backward reaction $k_{HF\to MF^*}$ of the dynF population depends only moderately on [NS] concentration, since $O_{op}^{HF}$ and $H_{op}^{HF}$ have similar rates for the closing transition. In conclusion, we can distinguish these two transitions because they have distinct characteristic equilibrium constants (Fig. 4f, Supplementary Fig. 2b, c), so that the joint observables, the average FRET efficiency and the relaxation time, shift as function of the total nucleosome concentration (Fig. 4e).

**The dimer:tetramer interface regulates the MF\*- HF transition.** What is the driving force behind the $MF^* \rightleftarrows HF$ transitions in nucleosomes? To answer this question, we used smFRET to analyze the stability of 20 pM $I_\beta I_\alpha$ nucleosome construct by varying [NaCl] from 5 to 1000 mM (Fig. 5a). Below 600 mM [NaCl], DNA binding was intact for the majority of nucleosomes states. At higher salt, we observed a steep decrease for the MF- and a correlated increase for LF-species fraction, suggesting gross DNA unwrapping, on a time scale of minutes to hours (Supplementary Fig. 3, Supplementary Table 6). Around 800–900 mM [NaCl] the majority of nucleosomes was found in a dynamic transition between MF\* and HF. The interdye distances did not vary significantly with salt, indicating that the structures of MF and LF remained unchanged (Supplementary Fig. 2d). Above 1 M NaCl, almost all nucleosomes were dissociated. The rate constants of the $MF^* \rightleftarrows HF$ transition changed with increasing salt concentration (Fig. 5b); $k_{MF^*\to HF}$ decreased while the back reaction rate constant increased (Fig. 5b), shifting the dynamic equilibrium between the two populations towards the closed configuration (Fig. 5c).

Previously, we suggested a disruption of the dimer:tetramer interface as an intermediate step in disassembly[14]. If this process was responsible for HF formation, mutations at the interface

should affect MF\*⇌HF dynamics. To test this hypothesis, we introduced the point mutation H2A-R81A in the dimer:tetramer interface at the position which is important for nucleosome stability[39,40] (Fig. 6a) and studied its impact on the dynF population. We first confirmed that the mutation did not affect nucleosome positioning and quality of reconstitution (Supplementary Fig. 4a), or fluorophore mobility as verified by MFD (Supplementary Fig. 4b, c). Salt-induced nucleosome disassembly in ensemble FRET (Fig. 6b) confirmed that mutated nucleosomes were significantly less stable than wt nucleosomes: $c_{1/2}(I_\beta I_\alpha\text{-wt}) = 783 \pm 5$ mM, $c_{1/2}(I_\beta I_\alpha\text{-mut}) = 618 \pm 7$ mM (standard errors, $N = 6$). Mutated nucleosomes were also more sensitive to low [NS] concentration (20 pM) so that we used 100 pM total nucleosome concentration for the smFRET measurements to map the structural heterogeneities of H2A-R81A and wt nucleosomes between 150 and 1000 mM NaCl. While both constructs behaved similarly at 150 mM salt (Supplementary Fig. 4d, e), even moderately higher NaCl concentrations revealed considerable differences (Fig. 6c, d). H2A-R81A mutation promoted conversion of the MF species into dynF without increasing the LF species fraction on the time scale of 1000 s. Gross dissociation and an increase in LF occurred only above 600 mM NaCl (Supplementary Fig. 4f–i). Finally, kinetic rates for the $MF^* \rightleftarrows HF$ transition were determined by dynPDA. Again, the H2A-R81A mutation did not alter kinetic rates at 150 mM NaCl, but caused significantly increased kinetic rates at intermediate ionic strength with a doubled opening rate constant $k_{MF^*\to HF}$ (Supplementary Fig. 4j, k).

Taken together, the data on $I_\beta I_\alpha$-mut nucleosomes support our hypothesis that partial opening of the dimer:tetramer interface is responsible for the $MF^* \rightleftarrows HF$ transitions. Its dependence on nucleosome concentration (Fig. 4f) indicates that the histone core contains at least one dimer in both dynF states.

**Geometric structural models of the HF and MF\* species.** Finally, we tried to connect the kinetic scheme with the FRET species, MF\* and HF, with structural candidates for sub-nucleosomal particles that can open up at the dimer:tetramer interface by an angle $\theta$ using a simplified geometric model of the nucleosome (Supplementary Note 3). The model was

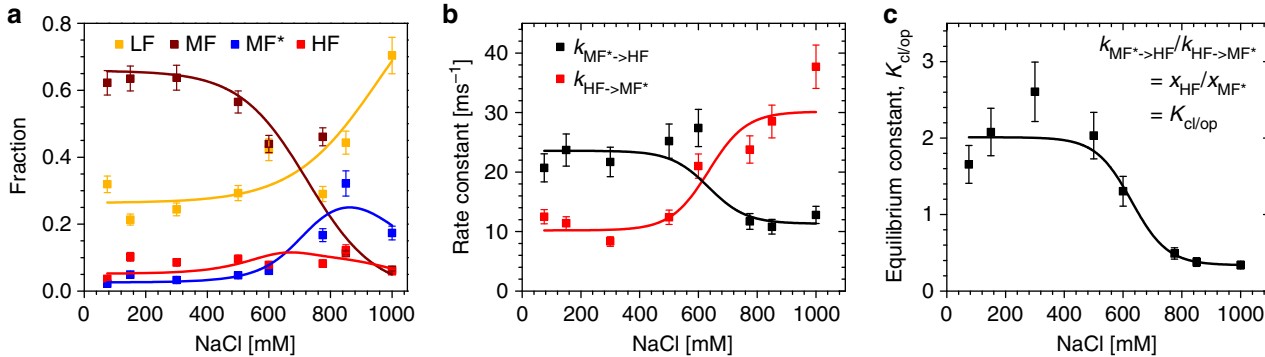

**Fig. 5** Kinetic parameters of $I_\beta I_\alpha$-wt nucleosomes as function of NaCl concentration. The dependence of fractions of the four FRET species on the NaCl concentration were determined from single molecule measurements of 20 pM $I_\beta I_\alpha$-wt nucleosomes by dynPDA. The transitions are described by weighted fits to a sigmoidal function (Supplementary Note 5, Supplementary Equation 36). **a** Salt-dependence of the static fractions $x_i$ of $i =$ LF, MF, MF\*, and HF. Intact nucleosomes (MF) are dominant at low salt concentrations but decrease rapidly above 600 mM NaCl in favor of LF and MF\*. The fraction of nucleosomes that was found in a dynamic transition between MF\* and HF peaks around 800–900 mM NaCl. Global fit with $x_{MF^*} = (1 - x_{LF} - x_{MF})/(1 + K_{cl/op})$ and $x_{HF} = (1 - x_{LF} - x_{MF})^* K_{cl/op}/(1 + K_{cl/op})$. The salt-dependent equilibrium constant $K_{cl/op}$ was obtained from the fit in **c**. $c_{1/2}$ (MF) = 725 ± 25 mM, $c_{1/2}$ (LF) = 957 ± 41 mM. **b** Salt-dependent kinetic rates for the $MF^* \rightleftarrows HF$ transition. Transitions from MF\* to HF are suppressed at higher NaCl concentrations, while the back reaction from HF to MF\* is promoted by increasing salt. Global fit with $c_{1/2} = 635 \pm 60$ mM. **c** Equilibrium constant $K_{cl/op}$ as obtained from the rates in **b**: $K_{cl/op} = k_{MF^*\to HF}/k_{HF\to MF^*} = x_{HF}/x_{MF^*}$. $c_{1/2}(K_{cl/op}) = 629 \pm 34$ mM. Error bars are standard errors from at least three independent measurements

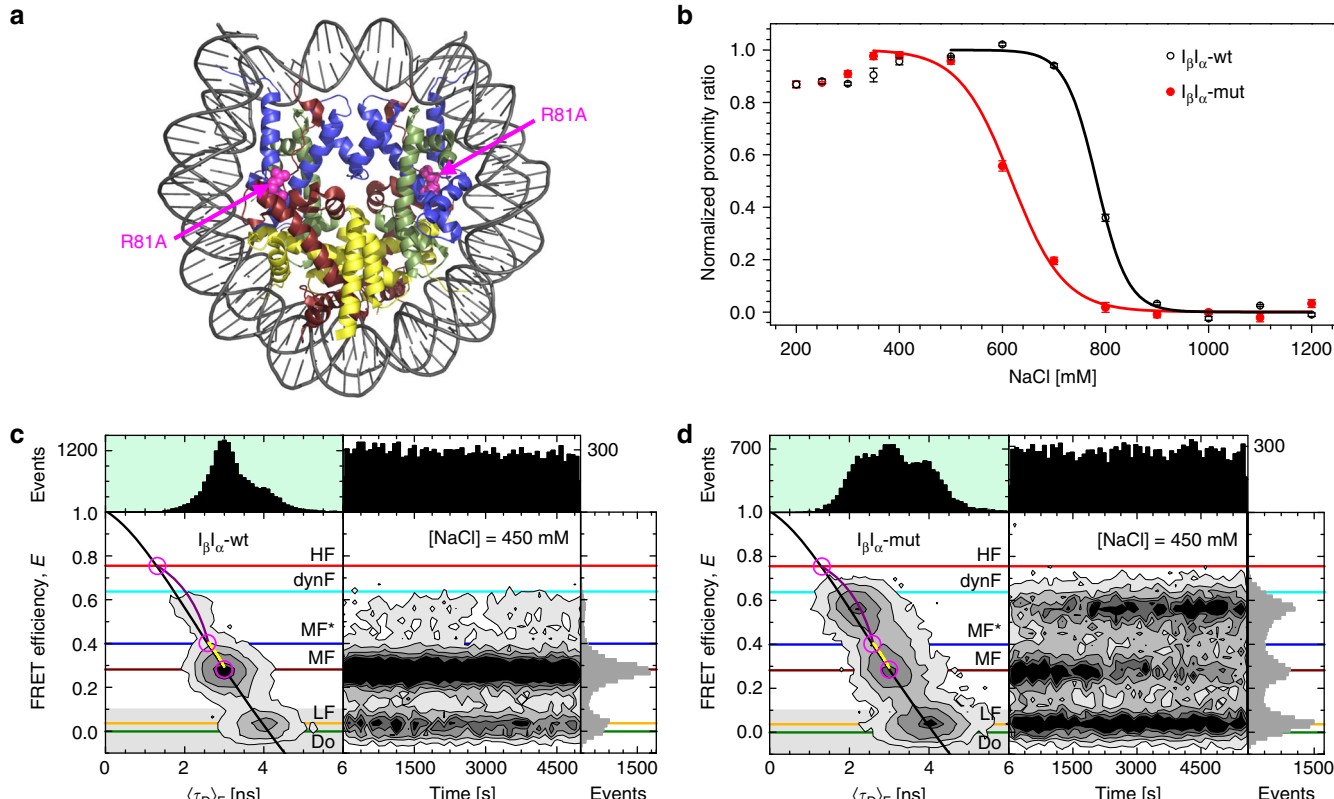

**Fig. 6** Effect of the point mutation R81A in H2A on nucleosome dynamics and disassembly. **a** Crystal structure (3LZ1) with the H2A-R81A mutation (shown in a space filling magenta representation) that modifies the charge at the dimer:tetramer interface. Histones are shown in blue (H3), pale green (H4), ruby (H2A) and yellow (H2B). **b** Ensemble FRET analysis of salt-induced disassembly of $I_\beta I_\alpha$-wt and $I_\beta I_\alpha$-mut nucleosomes. Nucleosomes carrying mutated H2A are more susceptible to elevated ionic strength than their non-mutated counterpart; $c_{1/2}(I_\beta I_\alpha$-mut) is about 165 mM lower than $c_{1/2}(I_\beta I_\alpha$-wt). Average values from 6 independent measurements were: $c_{1/2}(I_\beta I_\alpha$-wt) = 783 ± 5 mM and $c_{1/2}(I_\beta I_\alpha$-mut) = 618 ± 7 mM. **c** MFD-smFRET analysis of 100 pM $I_\beta I_\alpha$-wt nucleosomes at 450 mM NaCl. The majority of nucleosomes are found in MF; all three subspecies (LF, MF and dynF) remained stable over time. **d** MFD-smFRET analysis of 100 pM $I_\beta I_\alpha$-mut nucleosomes at 450 mM NaCl. Over time, a significant portion of intact nucleosomes (MF) transit into the dynamic FRET state. The fraction of fully open nucleosomes (LF), however, did not change over time. All colored lines in panels **c** and **d** match the lines in Fig. 4a, c. Error bars are standard errors from three replicates

parameterized based on the crystal structure (PDB ID: 3LZ1) and the simulation of accessible fluorophore space in $I_\beta I_\alpha$ nucleosomes to determine distances between the mean positions (mp) of donor and acceptor, $R_{mp}$[26] (Supplementary Note 3), which were subsequently converted to the experimentally observable $\langle R_{DA}\rangle_E$. The computed distance for the intact nucleosome is $\langle R_{DA}\rangle_E = 61$ Å, in excellent agreement with our experimental data. Distances in hexasomes were computed by removing one of the dimers and allowing DNA to exit tangentially from the disrupted dimer:tetramer interface (Fig. 7). This can be justified since the distance between tetramer end and dye on the free DNA segment is less than 20 bp, much smaller than its persistence length. Likewise, the z-pitch of the free DNA segment was assumed unchanged. For hexasomes missing the dimer at the β-side we computed $\langle R_{DA}{}^\beta\rangle_E = 56$ Å, while dimer removal at the acceptor site (α-side) yielded $\langle R_{DA}{}^\alpha\rangle_E = 58$ Å. Recall that the latter is the dominant hexasome species in MF* due to asymmetric dimer eviction. The small difference in distance between the intact nucleosome and the hexasome species agrees with the slightly smaller donor fluorescence lifetime measured for MF* in our MFD experiments.

For the modelling we considered (i) single opening of the nucleosome at one of the dimers (wine and orange lines), (ii) opening of the octasome at both dimer interfaces (light blue line), and (iii) opening of a hexasome that is already partially open (gray and black lines). The computed interdye distances for all

potentially dynamic species as a function of $\theta$ are shown in Fig. 7 together with the average experimental distances for both transient FRET species HF and MF* (red and blue ribbons). In the intact nucleosome $\langle R_{DA}\rangle_E$ depends only weakly on $\theta$ (wine and orange curves) and crosses the MF* values at about 15–20°. If only one histone dimer detaches, one would need $\theta > 70°$ to reach the HF interdye distance. Such a large angle would require sharp overbending of DNA, which is energetically unfavorable. However, a simultaneous opening of both interfaces in octasomes (light blue curve) or in hexasomes (black and gray curves) by ≈20° would already suffice to produce a mean dye distance comparable to $\langle R_{DA}{}^{HF}\rangle_E$ (Fig. 7). This corroborates our idea, that HF might be an open octasome $O_{op}^{HF}$ or hexasome $H_{op}^{HF}$ state and MF* is a closed octasome $O_{cl}^{MF*}$ or hexasome $H_{cl}^{MF*}$ state.

Furthermore, we need to rule out other mechanisms that could result in similar FRET changes. Extended DNA breathing for example would not affect the mean dye distance until a significant portion of DNA is unpeeled. This would increase donor mobility, which we did not observe in our anisotropy studies (Supplementary Table 4 and Supplementary Note 6). Nucleosome gaping - an out-of-plane motion of the two DNA turns - would increase the mean dye distance, contrary to the decrease observed in HF. Moreover, the characteristic time scale for this process is a few minutes[29], orders of magnitude slower than the MF* ⇌ HF transition.

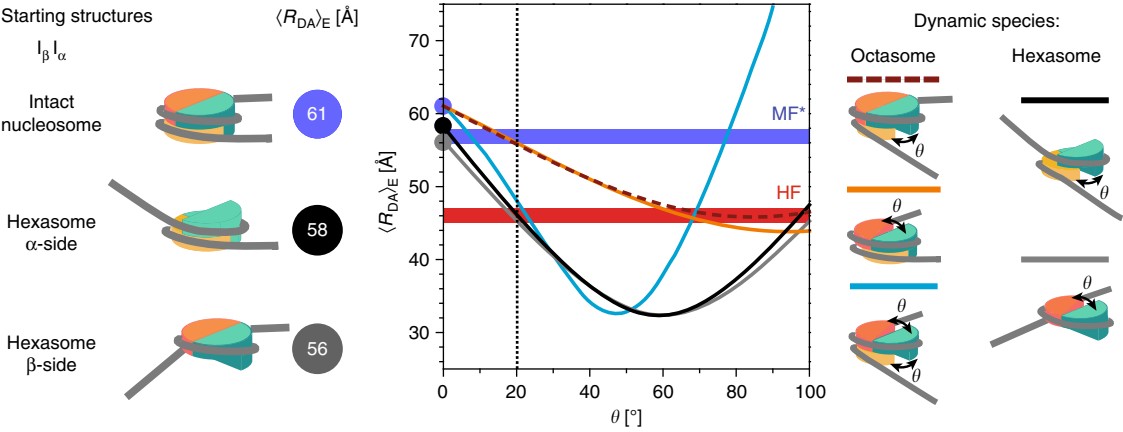

**Fig. 7** A geometric model for nucleosome dynamics. All details on the parameterization of the intact nucleosome and definitions of relevant parameters and the computer code are given in Supplementary Note 3. The (H3-H4)$_2$ tetramer is shown in turquoise, while H2A-H2B heterodimers are shown in orange (α-side) and dark yellow (β-side). The interdye distances ⟨$R_{DA}^{MF*}$⟩$_E$ are computed for the following three starting structures (opening angle $θ = 0°$): the intact nucleosome and both possible hexasomes with an open α-side and β-side, respectively. We compute the geometry of potentially dynamic nucleosome species as a function of the opening angle $θ$ (between the H2A-H2B dimer and the (H3-H4)$_2$ tetramer) for five scenarios dimer:tetramer opening: (i, ii) intact nucleosomes (octasomes), where opening occurs either only on the α-side (orange line) or only on the β-side (wine dashed line), (iii) intact nucleosomes (octasomes) with simultaneous opening on both sides with equal angles (cyan line) and (iv, v) hexasomes that are missing the dimer at the α-side (black line) or at the β-side (gray line) with additional opening on the other so far closed side. The blue and red ribbons represent the ranges of experimentally determined average interdye distances for MF* (⟨$R_{DA}^{MF*}$⟩$_E$ = (57 ± 1) Å) and HF (⟨$R_{DA}^{HF}$⟩$_E$ = (46 ± 1) Å), respectively. Our model predicts an simultaneous and equal opening by ≈20° within a hexasome or an octasome, while rather unrealistic angles greater than 70° would be required for nucleosome species with only one open dimer:tetramer interface

**Intermediates of nucleosome disassembly**. Considering the stepwise nucleosome destabilization with increasing NaCl concentration, our results obtained by four different label positions (summarized in Supplementary Table 5) and the H2A-R81A mutant (Fig. 1) led to a coherent picture of the structural features of the corresponding nucleosome states. In the dynamic octasome the DNA was not significantly unwrapped and the first salt-induced event was the dimer release on the α-side ($c_{1/2}$(H2B-Dy$_α$(D$_α$A)) = 546 ± 39 mM, Fig. 2c) that resulted in the formation of dynamic hexasomes as the majorly populated nucleosome state. Further NaCl increase led to a stepwise unwrapping of the DNA ends: firstly, on the α-side ($c_{1/2}$(E$_α$Dy$_β$) = 631 ± 6 mM, Fig. 3) with a simultaneous change of the equilibrium constant, $K_{cl/op}$, for the exchange between closed and open nucleosome species (cl/op) favoring the more stable closed state H$_{cl}^{MF*}$ ($c_{1/2}$($K_{cl/op}$) = 629 ± 34 mM, Fig. 5c); secondly, on the β-side ($c_{1/2}$(E$_β$Dy$_α$) = 694 ± 12 mM, Fig. 3). Next, DNA opened up also within the hexasome (($c_{1/2}$(I$_β$I$_α$) = 783 ± 5 mM, Fig. 6b) and the tetrasome with extended DNA was formed by release of the second dimer ($c_{1/2}$(H2B-Dy$_α$(D$_β$A)) = 866 ± 33 mM, Fig. 2). Notably, most salt-induced nucleosome transitions have broad half-widths $b$ in the range of 100–150 mM, (Supplementary Table 5) which indicates heterogeneity with multiple intermediates.

Combining the dynamic and structural features of the nucleosome states revealed by our smFRET studies, we suggest a disassembly pathway of the intact nucleosome O$_{st}^{MF}$ with at least five major steps and seven characteristic intermediates (Fig. 8). Moreover, we determined all rate and equilibrium constants by detailed kinetic analysis of the MFD data (relaxation times and dynPDA species fractions, see Supplementary Note 2) for physiological conditions (150 mM NaCl) and compiled them in Table 1.

In step I, the dimer:tetramer interface slowly opens at the weaker α-side into a metastable intermediate O$_{cl}^{MF*}$ in rapid equilibrium with an open conformation O$_{op}^{HF}$ through reversible disruption of the second dimer:tetramer interface (step II, tens of

microseconds). There are several corroborating observations that the initial opening of the intact nucleosome (step I) is slow (>3 ms): (i) simulations for reproducing the FRET populations in the MFD diagrams with both FRET-lines connecting the static octasome O$_{st}^{MF}$ and the dynamic octasome, O$_{op}^{MF*}$ and O$_{op}^{HF}$, indicate that the opening rates at the α-side must differ by factor >1000; (ii) this estimate agrees with a relaxation time of 5 ms reported by Lee et al.[18,41]. It is obvious that DNA breathing[5,42] is essential for the motions of the histone core, which explains the complex kinetics with several additional relaxations times between 1 and 100 μs that is revealed by further analysis of the MFD data using filtered Fluorescence Correlation Spectroscopy (fFCS) (Supplementary Fig. 5 and Supplementary Note 4). While dynPDA is most sensitive to the internal opening dynamics in the time range 15–50 μs, fFCS monitors all FRET fluctuations in I$_β$I$_α$ nucleosomes over four orders of magnitude in time (sub μs to ms). O$_{op}^{HF}$ is reminiscent of the "butterfly" structure, originally sketched in our previous work[14]. In step III the α-dimer dissociates, leading to a second intermediate, the dynamic hexasome with the states H$_{cl}^{MF*}$ and H$_{op}^{HF}$ (step IV) with slightly faster dynamics of the remaining dimer and partially unwrapped DNA (dominant species at 700 mM NaCl). The final disassembly (step V) is the transition into tetrasomes T$^{LF/NF}$ through loss of the second, the β-dimer. This last transition is slow (minutes to hours time scale) with rates comparable to those recently published by Plavner-Hazan et al.[43]. Note that at NaCl concentrations >800 mM tetrasomes are formed more easily (Supplementary Fig. 3). Tetrasome formation and subsequent tetrasome dissociation are both irreversible under single-molecule conditions (nM concentrations), in contrast to the stable equilibrium conditions at lower NaCl concentrations.

## Discussion

The study of nucleosomes by salt-induced destabilization monitored by FRET and other biophysical techniques on the single-molecule and ensemble level has been frequently applied to reveal their heterogeneity[11,12,14,17,36,42,44–53].

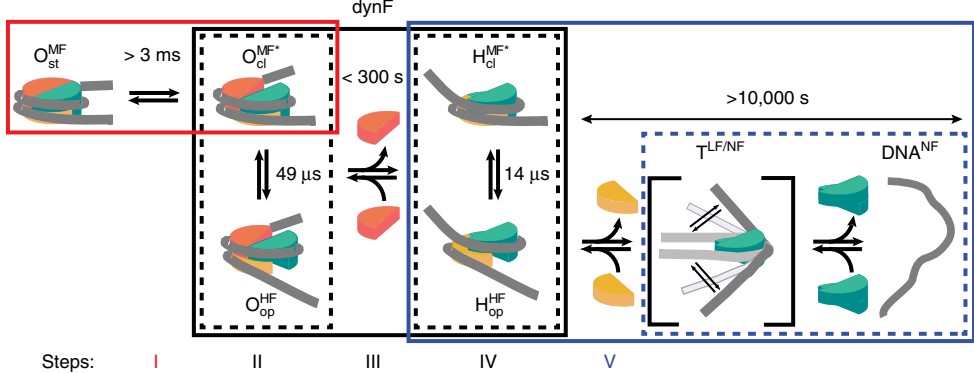

**Fig. 8** Full kinetic scheme for nucleosome disassembly. At 150 mM NaCl concentration nucleosomes disassembly proceeds through the sequential loss of H2A-H2B heterodimers as presented here with the corresponding relaxation times. The two heterodimers are shown in orange (α-side) and dark yellow (β-side), while the tetramer is shown in turquoise. We resolve five steps in our smFRET experiments: (I) Disassembly of the intact nucleosome $O_{st}^{MF}$ is predominantly initiated from the weaker binding α-side, where opening of the first dimer:tetramer interface precedes dimer eviction $O_{cl}^{MF*}$. (II–IV) The second dimer can reversibly detach from the tetramer once the first dimer:tetramer interface is broken $O_{op}^{HF}$. The rate constant for the second dimer opening, however, is smaller when the first dimer is still present, probably due to stabilizing interactions between the dimers. Once the first dimer is gone $H_{cl}^{MF*}$, reversible detachment $H_{op}^{HF}$ and subsequent loss of the remaining dimer proceed at a faster rate. (V) The resulting tetrasome can adopt variable DNA geometries with most conformations leading to very low or no FRET ($T^{LF/NF}$). Note that at higher salt concentration the DNA can fully dissociate from the remaining tetramer ($DNA^{NF}$). All corresponding kinetic and thermodynamic parameters are compiled in Table 1

### Table 1 Kinetic and thermodynamic parameters of the nucleosome disassembly at 150 mM NaCl[a]

| Step | Rate constants | | Equilibrium constants | | Relaxation times | | Method[b] |
|---|---|---|---|---|---|---|---|
| I | $k_{O(st)\to O(cl)}$ $k_{O(cl)\to O(st)}$ | $<0.02\times10^3$ s$^{-1}$ $<0.32\times10^3$ s$^{-1}$ | $K_{st/cl}^O$ | $0.06 \pm 0.01^c$ | $t_R^{O(st/cl)}$ | $>3$ ms$^c$ | a, b (Figure 4) |
| II | $k_{O(cl)\to O(op)}$ $k_{O(op)\to O(cl)}$ | $(13 \pm 1)10^3$ s$^{-1c}$ $(8.0 \pm 0.6)10^3$ s$^{-1}$ | $K_{cl/op}^O$ | $1.6 \pm 0.2$ | $t_R^{O(cl/op)}$ | $(49 \pm 2)$ µs | a (Figure 4, Supplementary Fig. 4) |
| III | $k_{O\to H}$ $k_{H\to O}$ | $>4\times10^{-3}$ s$^{-1}$ $>10^8$ M$^{-1}$ s$^{-1}$ | $K_{dis}^{O/H}$ | $(39 \pm 12)$ pM$^c$ | $t_R^{O/H}$ | $<300$ s$^{c, d}$ | a, c (Figure 4, Supplementary Fig. 3) |
| IV | $k_{H(cl)\to H(op)}$ $k_{H(op)\to H(cl)}$ | $(54 \pm 5)10^3$ s$^{-1c}$ $(17 \pm 2)10^3$ s$^{-1c}$ | $K_{cl/op}^H$ | $3.3 \pm 0.5$ | $t_R^{H(cl/op)}$ | $(14 \pm 6)$ µs | a (Figure 4, Supplementary Fig. 3) |
| V | $k_{H(op)\to T}$ $k_{T\to H(op)}$ | $<(1.8 \pm 0.5)10^{-5}$ s$^{-1c}$ $<10^7$ M$^{-1}$ s$^{-1c}$ | $K_{dis}^{H/T}$ | $>2$ pM | $t_R^{H/T}$ | $>10,000$ s$^{c, d, e}$ | c (Supplementary Fig. 2) |

[a]The reaction scheme and the nucleosome species are displayed in Fig. 8 with the following characterization: (O) octasome, (H) hexasome, (T) tetrasome; (st) static closed, (cl) dynamic closed, (op) dynamic open
[b]Methods: (a) dynPDA, (b) simulation, (c) trace analysis
[c]Directly determined parameters
[d]Estimated for [H2A-H2B] ≈10 pM
[e]Note that at NaCl concentrations >800 mM tetrasomes (Supplementary Fig. 3) are formed more easily. In the last step DNA dissociates irreversibly under single-molecule conditions

In recent smFRET experiments[43], salt-induced dissociation kinetics were measured on a minute-to-hour time scale at 5 and 300 mM NaCl and 3 pM nucleosome concentration. While these conditions spanned the physiological salt range, the kinetics could only describe the slow transition from the intact to the partly dissociated nucleosome (the $T^{LF/NF}$ state) without additional resolved intermediates. Recently, Chen et al. studied nucleosome disassembly kinetics in very high salt (up to 1.9 M NaCl) by stopped-flow with small-angle X-ray scattering (SAXS) and ensemble FRET detection. The development of the nucleosomal ensemble over time led to postulating unwrapping states with transition kinetics in the range of 50 milliseconds to seconds. Due to the reduced data quality in fast time-resolved SAXS, structural predictions from such experiments are necessarily limited in their precision. Therefore, we applied smFRET with MFD, dynPDA, fFCS and detailed analysis for potential structural species to gain an in-depth view of nucleosome disassembly at arbitrary salt concentrations and short time scales. We revealed complex kinetics with fast exchanging initially opened states (with a dominating

relaxation time of tens of microseconds), in which the nucleosome internal structure is loosened and the DNA is still wrapped but potentially breathing[5,42]. We suggest a multistep kinetic scheme that provides details of nucleosome assembly and disassembly and critically depends on the overall nucleosome concentration. In the following, we discuss four important observations.

1. Nucleosome dissociation begins with DNA unwrapping. Free in solution, the 601 nucleosomes preferentially open up from one side, where the DNA is more loosely bound. We show for hexasomes that further disassembly starts with sequence-dependent DNA unwrapping in the entry–exit region at the α-side and detect additional kinetic transitions (~0.1 ms) by fFCS. While asymmetric DNA unwrapping in surface-attached nucleosomes under tension was reported by Ngo et al.[34], no direct evidence existed for nucleosomes without external tension in solution. Ngo et al. reported that the DNA half whose central region contains multiple TA-steps at ~10 bp spacing (our "β-side") is more resilient against mechanical unwrapping than the weaker binding DNA side. This agrees with our observations that

local flexibility of DNA controls nucleosome stability directly, even without external tension.

2. Nucleosome dynamics is linked with H2A-H2B dimer exchange. The correlation of DNA stiffness and H2A-H2B dimer eviction supports earlier in vitro observations that the transcription barrier formed by the nucleosome is polar[54]. However, this hypothesis is still controversial: a recent study[55] did not find an orientation-dependent difference in transcription rates through 601 nucleosomes, implying that factors beyond DNA sequence must play a role[56,57]. The fast dynamic transition shown here agrees with our earlier finding[14] that before dimer eviction, the dimer:tetramer interface is disrupted transiently. A cleft, some nm in size, within the histone core could then serve as a recognition motif for typical nuclear proteins. DNA-histone interactions are clustered in 14 discrete regions along the superhelical DNA path, where the average binding energy per cluster only slightly exceeds thermal energy[2,58]. In an octasome without external tension, DNA would unpeel by successive disruption of DNA-histone interactions from the DNA ends, exposing DNA inside the nucleosome much less frequently than in the entry/exit region[6]. After opening of the dimer:tetramer interface, as supposed in our model, DNA-histone contacts adjacent to the interface are no longer shielded from disruption, offering much easier access.

3. H2A-H2B dimers are step-wisely released. Earlier on we had proposed a step-wise disassembly through loss of H2A–H2B dimers[12], but the lack of analytical tools at this time precluded the detailed kinetic and structural finding that the dimer:tetramer interface is crucial for nucleosome stability and it must already open reversibly[14] in the dynamic octasome to prime the system for subsequent dimer eviction. Considering a $\approx 20°$ opening of the intact nucleosome (species MF), our geometric model predicts a distance change of $\approx 4$ Å ($\Delta E \approx 0.1$) that approximately corresponds to the FRET efficiency of MF* (Fig. 7).

Interactions between H2A-H2B dimers through hydrogen bonds between their L1 loop domains[1] stabilize the nucleosome, increasing the energetic barrier for the opening of the first dimer. Our structure-based finding that the initial opening might be significantly slower for the first than for the second dimer, nicely agrees with our observed millisecond and microsecond dynamics for the corresponding process (steps I and II $\left( O_{cl}^{MF*} \rightleftarrows O_{op}^{HF} \right)$ and IV in Table 1, respectively). Earlier ensemble FRET experiments suggested such a cooperativity[59], but by monitoring the dynamic octasome and hexasome intermediates it could be detected in this study.

Arimura et al. reported SAXS data that suggested an overall similar static core structure of hexasomes and intact nucleosomes[60]. Here, however, we found the hexasome state as part of the dynamic FRET population. Hexasomes are also highly dynamic entities undergoing rapid transitions (tens of microseconds) between open and closed histone core conformations. To our best knowledge, this constitutes the first report of such fast dynamics in hexasomes.

We finally note that the energetic barrier for the release of the first heterodimer could in principal also be lowered by nucleosome gaping, where the two (H2A-H2B)/(H3-H4) halves of the octamer transiently open-up in a clam-shell like motion[28,29]. Reversible opening of the dimer:tetramer interface could well occur in the "gaped" conformation, after disruption of the stabilizing interactions between the L1 loops. This is, however, not a likely candidate for MF* and HF, since the preceding MF-MF* transition exhibits strong concentration dependence, that is indicative for a dissociation process. Gaping as described by Ngo et al.[29], on the other hand, would not depend on nucleosome concentration.

4. Tetrasome disassembly. Finally, the second histone dimer dissociates, followed by full opening of the DNA and (H3-H4)$_2$ tetramer release. It is noteworthy that in the studies of[14] we observed full tetramer release almost simultaneously with full DNA unfolding already at 1.2 M salt[14], while Chen et al.[30] used 1.9 M salt to reach this limit. This could imply different opening mechanisms in high and low salt or simply caused by the non-equilibrium conditions inherent to stopped flow experiments reported in reference [30].

In conclusion, our results provide important insights into mononucleosome dynamics which could be more representative for terminal[61] than internal sites[20] in nucleosome arrays used as models for chromatin fibers[62]. MFD enabled us to observe octasome and hexasome dynamics in reversible opening of the dimer:tetramer interface. Further analysis of the concentration and salt dependencies allows us to postulate a minimum kinetic scheme for the nucleosome structure opening through at least seven species from the compact mononucleosome to the free DNA. Moreover, the methodology developed in this work paves the way to systematic investigations of the role of histone tail modifications and interactions in chromatin structure, enabling us to develop a detailed dynamic picture of this central biological entity.

## Methods

**Preparation of mononucleosomes.** Mononucleosomes were made from 170 bp DNA fragments containing the Widom 601 positioning sequence[32] and Xenopus laevis recombinant histones. DNA was prepared by PCR with fluorescent primers (IBA) and purified on Gen-Pak FAX HPLC (Waters). Recombinant wild type (wt) histones were expressed in E. coli bacteria and purified by gel filtration[42]. For some experiments we mutated histone H2A, replacing arginine at position 81 with alanine (H2A-R81A). H2B-T112C mutants for histone dimer labeling were purchased from Planet Protein (Colorado State University).

Histone octamers were prepared by mixing histones in unfolding buffer (7 M Guanidine) followed by dialysis against high salt (2 M NaCl in TE buffer: 10 mM Tris-HCl containing 0.1 mM EDTA, pH = 7.5). Octamers in best proportion were selected after size exclusion FPLC (Superdex 200HR 10/10) and Triton X-100/ acetic acid/urea (TAU) gel analysis.

In single color excitation experiments, FRET donor and acceptor were Alexa488 (A488) and Alexa594 (A594). In MFD-PIE, Cy5 was used as acceptor. The acceptor was attached to the DNA through amino-C6 linker either internally about half a turn away from the dyad axis (I$_\beta$) or near the dyad axis at either α- or β-side (Dy$_\alpha$ or Dy$_\beta$). Note that according to the nucleosome crystal structure the dyad is localized at around – 6 bp of our sequence[63]. The donor was attached to DNA via amino-C6 linker at one or the other end (E$_\alpha$ or E$_\beta$), at an internal position about half a turn away from the dyad axis (I$_\beta$) or bound to histone H2B at position 112 using the H2B-T112C mutant[14]. Five types of nucleosomes were prepared for FRET measurements (Table 2, see also Supplementary Methods):

---

**Table 2 Notation of nucleosome constructs**

| Notation | Sample description |
|---|---|
| I$_\beta$I$_\alpha$-wt | DNA labeled at +41 (A488) and −53 (A594) and wt octamers |
| I$_\beta$I$_\alpha$-mut | DNA labeled at +41 (A488) and −53 (A594) and octamers containing H2A-R81A mutated histones |
| E$_\beta$Dy$_\alpha$ | DNA labeled at 5′ end β (E$_\beta$) with A488 and at position −15 (close to the dyad) on the other strand (Dy$_\alpha$) with A594, plus wt octamers |
| E$_\alpha$Dy$_\beta$ | Analogous to nucleosome E$_\beta$Dy$_\alpha$ with DNA label positions on the complementary strands (A488 at 5′ end α, A594 at + 15) |
| H2B-Dy$_\alpha$ | DNA labeled at -15 (near the dyad) with Cy5 and octamers containing H2B-A488 |

In $I_\beta I_\alpha$ constructs the dyes were placed at the heterodimer-binding region of nucleosomal DNA, such that their local environment is sensitive to the presence or absence of H2A-H2B heterodimers.

DNA labeling efficiency was >95% as measured by absorption spectroscopy and by single-molecule fluorescence with alternating laser excitation. Labeling efficiency for histone H2B was ~45% as determined from TAU gel analysis and spectroscopic measurements. Nucleosomes were prepared by mixing DNA and histone octamers in TE buffer at high salt (2 M NaCl), then slow salt dialysis (mini-dialyzing tubes, cut off 7 kDa, Pierce) down to 5 mM. Molar DNA:Octamer ratios in the range 1:1.5 – 1:1.8 were adjusted such that <5% free DNA was visible after reconstitution. Donor-only labeled nucleosomes, free double labeled DNA and donor- or acceptor-only labeled DNA were prepared for control experiments from the same sources. For some experiments the overall nucleosome concentration was adjusted by adding unlabeled nucleosomes prepared with the same octamers. The quality of nucleosome reconstitution was tested by absorption spectroscopy for detecting possible aggregates and by PAA gel electrophoresis (6%, 60:1 acrylamide: bisacrylamide). Residual aggregates were removed by centrifugation (Eppendorf Centrifuge 5417 R) at 10,000 rpm and 10 °C for 10 min. Ensemble fluorescence anisotropy of the attached dyes was below $r = 0.2$ in DNA and nucleosome samples, indicating sufficient dye mobility. Nucleosomes with concentrations above 100 nM were stable at 4 °C for several weeks.

**Ensemble FRET measurements**. Ensemble FRET experiments were performed on the microplate reader, Typhoon multimode imager[36]. A freshly cleaned and passivated 384-well microplate was loaded with 300 or 600 pM labeled nucleosomes that had been incubated for 60 min at NaCl concentrations ranging from 5 to 1200 mM. To quantify FRET, the donor and acceptor fluorescence were recorded at 488 nm excitation using bandpass filters provided in the instrument (donor channel ($I_D^{Dex}$): 520BP40, acceptor channel ($I_A^{Dex}$): 610BP30). An additional scan of the acceptor signal at 532 nm excitation ($I_A^{Aex}$) was used to account for direct acceptor excitation. All control samples (buffer only, donor-only, acceptor-only and double labeled DNA fragments with no FRET) were measured in parallel to the actual nucleosome samples. For each sample, the proximity ratio ($P$) was computed from the measured fluorescence intensities considering calculated correction factors and averaged over three independent wells:

$$P = \frac{(I_A^{Dex})_{corr}}{(I_D^{Dex})_{corr}+(I_A^{Dex})_{corr}} \tag{1}$$

where $(I...)_{corr}$ represent corrected fluorescence intensities as described in Supplementary Note 5.

**Data analysis**. Ensemble $P$ data were normalized to a scale of 0% (no FRET) to 100% (maximum FRET) and approximated by a sigmoidal function (Supplementary Note 5, Supplementary Equation 36) to obtain the salt concentration at which the proximity ratio dropped to half of its maximal value ($c_{1/2}$ value). This value was taken as a measure for the stability of the corresponding FRET pair[36].

**Single-molecule FRET experiments**. MFD experiments were carried out with a confocal epi-illuminated setup based on an Olympus IX70 inverted microscope as described in refs. [31,64–67]. In brief, a linearly polarized pulsed 495 nm diode laser (LDH-D-C 495, PicoQuant; 36 μW in the sample) operating at 64 MHz was focused into the sample solution by a 60×/1.2NA water immersion objective (UPLAPO 60×, Olympus, Germany). The photon train was divided into its parallel and perpendicular components by a polarizing beamsplitter cube (VISHT11, Gsänger) and then into spectral ranges below and above 595 nm by a dichroic beamsplitter (595 LPXR, AHF). Additionally, red (HQ 645/75 nm for Alexa594) and green (HQ 533/46 nm for Alexa488) bandpass filters (both provided by AHF) in front of the detectors ensured that only fluorescence from the acceptor and donor molecules were registered, while residual laser light and Raman scattering from the solvent were blocked. Photons were detected by four avalanche photo-diodes (green channels: τ-SPAD-100, PicoQuant; red channels: SPCM-AQR-14, Perkin Elmer). The signals from all detectors were guided through a passive delay unit and two routers to two synchronized time-correlated single photon counting boards (SPC 132, Becker & Hickl) that were connected to a PC. The high frequency output of the diode laser serves as the common time base for synchronization (TCSPC stop signal) and the macro time counter for both SPC-132 counting boards. Bursts of fluorescence photons are distinguished from the background of 0.3–0.4 kHz by applying certain threshold intensity criteria[64]. For analysis, several parameters including fluorescence lifetime, anisotropy, and FRET efficiency were computed per burst to classify the molecules according to multidimensional relations between these parameters. Further details are given in Supplementary Methods.

MFD measurements with pulsed interleaved excitation (PIE) were performed employing a setup similar to the one described above. In PIE measurements[22], donor and acceptor are sequentially excited by rapidly alternating laser pulses. MFD can be performed on both dyes, allowing computation of the donor-acceptor ratio (stoichiometry, $S_{PIE}$) for each particle. Excitation is achieved using 485 nm and 635

nm pulsed diode lasers (LDH-D-C 485 and LDH-P-C-635B, respectively; both PicoQuant, operated at 32 MHz) on an Olympus IX71 microscope. Laser power in the sample was 34 and 7 μW, respectively. After separating the fluorescence signal according to color and polarization, each of the four channels was split again using 50/50 beam splitters in order to get dead time free fFCS curves, resulting in a total of eight detection channels. Detector types for green and red channels were the same as described above, bandpass filters were HQ 520/35 and HQ 720/150 (both AHF) for green and red channels, respectively. The detector outputs were recorded by a TCSPC module (HydraHarp 400, PicoQuant) and stored on a PC. Analysis of the single photon data stream was done using threshold criteria[22].

**Measurement conditions for ensemble and smFRET experiments**. For experiments nucleosomes were freshly diluted into 0.02 μm-filtered measuring buffer (TE containing NaCl as noted), 1 mM ascorbic acid (Sigma-Aldrich) and 0.01% Non-idet P40 (Roche Diagnostics) and unlabeled nucleosomes as noted. 50–100 μl of nucleosomes were placed into 384-well microplates (SensoPlate Plus, Greiner Bio-One) that had been passivated with Sigmacote (Sigma-Aldrich). In ensemble FRET experiments the passivated microplate was loaded with 300 or 600 pM labeled nucleosomes that had been incubated for 60 min at NaCl concentrations ranging from 5 to 1200 mM. If not otherwise stated, in single-molecule FRET experiments 20 pM of labeled nucleosomes were used. Where necessary, an appropriate amount of unlabeled nucleosomes was then added to adjust overall nucleosome concentration. Data were taken for at least 90 min per sample. Series of measurements, e.g. salt titration or nucleosome titration, were repeated and results of the respective analyses averaged. Due to slight day-to-day variations in the sample preparations only complete series were used for averaging. Displayed error bars reflect these variations.

**Determination of FRET-based interdye distances**. To confirm our FRET-based identification of structural substates for the construct H2B-Dy$_\alpha$, we measured donor mobility via its burstwise anisotropy $r_D$ (Supplementary Fig. 1b, Supplementary Table 4, Supplementary Note 6) to validate the used $R_0$, assuming a FRET orientation factor $\kappa^2 = 2/3$ for the isotropic average. Each population was well described by a single donor anisotropy value (dynF ($r_D = 0.135$), MF ($r_D = 0.114$), and LF ($r_D = 0.080$)), decreasing with increasing lifetime. The anisotropies of all FRET species are in agreement with high mobility (average rotational correlation time $\rho \approx 1$ ns), so that $\kappa^2 = 2/3$ can be safely assumed.

For the five nucleosome constructs we employed two FRET pairs with the following Förster radii: $R_0$(Alexa488 - Cy5) = 52.0 Å and $R_0$(Alexa488 - Alexa594) = 55.6 Å. The crystal structure of mononucleosome (PDB ID: 3LZ1) was used as template for the interdye distance prediction (for details (geometric description, computer code and comparison with earlier work[11] see Supplementary Note 3). The dye distribution was modeled by the accessible volume approach (AV). Here donor and acceptor fluorophores are approximated by a sphere with an empirical radius of $R_{dye}$, where the central atom of the fluorophore is connected by a flexible linkage of a certain effective length $L_{link}$ and width $w_{link}$. All geometric parameters for the dyes were: Alexa488 and Alexa594: $L_{link} = 20$ Å, $w_{link} = 4.5$ Å, $R_{dye} = 1.5$ Å; Cy5: $L_{link} = 22$ Å, $w_{link} = 4.5$ Å, $R_{dye} = 3.5$ Å.

For better direct comparability, we convert average distances from different data-sources, i.e. mean distance, $\langle R_{DA} \rangle$, from fluorescence lifetime analysis; FRET averaged distance, $\langle R_{DA} \rangle_E$, from intensity analysis by PDA and distances between the mean positions of donor and acceptor, $R_{mp}$, from geometric modeling (Supplementary Note 3) to $\langle R_{DA} \rangle_E$; e.g., $\langle R_{DA} \rangle$ is converted to FRET-average mean distance $\langle R_{DA} \rangle_E$ by the following polynomial: $\langle R_{DA} \rangle_E = 15.3230 + 0.592830 \langle R_{DA} \rangle + 0.00175370 \langle R_{DA} \rangle^2 - 0.0000032013 \langle R_{DA} \rangle^3$. The coefficients of the other polynomials are compiled in Supplementary Table 1.

**Code availability**. Most general custom-made computer code is directly available from http://www.mpc.hhu.de/en/software. The source code for simulating the curves in Fig. 7 using the programming tool IGOR Pro (WaveMetrics, Lake Oswego, OR, USA) is provided in Supplementary Note 3. Additional computer code custom-made for this publication is available upon request from the corresponding authors.

## Data availability

Other data supporting the findings in this work are available from the corresponding authors upon reasonable request.

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

## Acknowledgements

We dedicate this paper to the memory of Jörg Langowski, a brilliant pioneer in studying chromatin in silico, in vitro, and in live cells. He had the courage to follow his heart and intuition in life and research. We thank Nathalie Schwarz for expert technical assistance in preparing the nucleosome samples and Mykola Dimura for AV simulations. This work was supported by DFG grants La 500/18-1 to J.L. and Se 1195/15-1 to C.A.M.S.; This research was also supported by the European Research Council through the Advanced Grant 2014 hybridFRET (671208) to C.A.M.S.

## Author contributions

A.G., K.L., and K.T. prepared samples. A.G. and K.L. performed and analyzed ensemble FRET experiments. A.G., K.L., K.T., J.L., S.F., and R.K. performed smFRET experiments. S.F., R.K., and C.A.M.S. analyzed smFRET data. R.K. performed kinetic analysis. A.G., S.F., K.L., and R.K. prepared the figures and discussed them with all authors. All authors participated in the writing of the manuscript. SI was collected by S.F., R.K., and A.G.; J.L., K.T., and C.A.M.S. supervised the project.

## Additional information

**Competing interests:** The authors declare no competing interests.

