## [Peer Review File · Nature Communications]

Reviewers' comments:

Reviewer #1 (Remarks to the Author):

This is a very nice manuscript that represents an outcome of applying several advanced fluorescence technologies developed by the Seidel lab over the years to a very interesting problem of nucleosome dynamics such as dimer opening and partial unraveling of DNA. It is impressive how much information they can extract from the single molecule fluorescence data, for example, the existence of four different noncanonical nucleosomal states with one dimer or the other dimer cracking open while keeping both dimers or only one dimer. And the exchange time scale is in the microseconds! The dimer on the less flexible side of the DNA opens up, and I do not think there is any previous indication of such asymmetric opening. The manuscript is pretty technically demanding to understand but the biological messages are clearly presented for those who do not wish to get into technical details. I have just a couple of minor suggestions.

1. "we hypothesize that the two MF' \rightleftharpoons HF transitions occur within the octasome and the hexasome. These transitions have to be fast and are coupled by the eviction/uptake of one heterodimer." I am not sure what is meant by "coupled by".
2. For the main data set acquired using I_{alpha}-I_{beta}, I cannot tell how they could deduce that the alpha side is opening up first.
3. When comparing their work with Ngo et al, they used the term 'perturbed' vs 'without external perturbation'. I know what they mean, i.e. with or without external tension applied. But in the current study, they use high salt concentration solution which might also be considered perturbation. I would suggest they change it to external tension. It is actually quite likely that tension applied puts the nucleosome in the more physiologically relevant state compared to high salt condition.

Reviewer #2 (Remarks to the Author):

The manuscript by Gansen, et al. reports multiparameter single molecule fluorescence studies of Cy3-Cy5 labeled nucleosomes. They results on hexasome dynamics and the that nucleosome undergo rapid unwrapping with the H2A-H2B heterodimer attached to the DNA. The multiparameter single molecule fluorescence studies are ideal for studying nucleosome dynamics. So, I was looking forward to reading about this work on using their multiparameter fluorescence detection (MFD). However, this manuscript is (at least for me) extremely difficult to read. It is full of abbreviations that are not defined and it assumes the reader is already familiar with the experimental design. While this reviewer is familiar with the authors previous work on MFD and nucleosome studies, I was continually trying to find definitions of all of their abbreviations and nomenclature. It was taking so long to review the manuscript, that I decided to stop and recommend that the authors rewrite the paper so it is readable and then resubmit the paper.

Reviewers' comments:

Reviewer #1 (Remarks to the Author):

This is a very nice manuscript that represents an outcome of applying several advanced fluorescence technologies developed by the Seidel lab over the years to a very interesting problem of nucleosome dynamics such as dimer opening and partial unraveling of DNA. It is impressive how much information they can extract from the single molecule fluorescence data, for example, the existence of four different noncanonical nucleosomal states with one dimer or the other dimer cracking open while keeping both dimers or only one dimer. And the exchange time scale is in the microseconds! The dimer on the less flexible side of the DNA opens up, and I do not think there is any previous indication of such asymmetric opening. The manuscript is pretty technically demanding to understand but the biological messages are clearly presented for those who do not wish to get into technical details. I have just a couple of minor suggestions.

1. "we hypothesize that the two MF* \rightleftharpoons HF transitions occur within

the octasome and the hexasome. These transitions have to be fast and are coupled by the eviction/uptake of one heterodimer." I am not sure what is meant by "coupled by".

The expression "coupled by" might have been a bit misleading (we mean here "due to" or "related"). We changed "coupled by" to "coupled with".

Furthermore, an explanation for our assumption can be found on page 8 (of the revised manuscript):

"Importantly, the average FRET efficiency $\langle E \rangle$ for the dynF peak decreased by 0.08 with increasing nucleosome concentration (Fig. 4E, top), while the relaxation time for the MF* \rightleftharpoons HF transition increased approximately by a factor of 3 (Fig. 4E, bottom).

This concentration dependence implies that the transition from MF* to HF involves an association/dissociation equilibrium, most likely the uptake/eviction of an H2A-H2B dimer. Even if this process were diffusion limited, this process would have relaxation times of ms or slower at sub-nanomolar concentrations. Thus, a single MF* \rightleftharpoons HF transition is not compatible with the observed fast microsecond kinetics."

2. For the main data set acquired using I_{alpha}-I_{beta}, I cannot tell how they could deduce that the alpha side is opening up first.

Reviewer 1 is right, it is not possible to deduce this from opening of construct I_{beta}I_{alpha}, but our conclusion was not based on this construct. Instead, data obtained with constructs E_{beta}Dy_{alpha} and E_{alpha}Dy_{beta}, unveiling different $c_{1/2}$ values for alpha- and beta-side, were used for this conclusion (can be found on page 5 of the revised manuscript). Moreover we added a section "Intermediates of nucleosome disassembly" (page 10 in the revised manuscript) where we summarize the nucleosome destabilization experiments with increasing NaCl concentration for all FRET construct and demonstrate that they lead to a coherent picture of the structural features.

"There we compared salt-dependent DNA unwrapping of the two nucleosome ends by multiplexed ensemble FRET³⁵ on constructs with double labeled DNA, E_{alpha}Dy_{beta} and E_{beta}Dy_{alpha} (Supplementary Methods 1). Labeled nucleosomes were diluted to 600 pM and incubated at different NaCl concentrations for 60 minutes. Both constructs behave similarly in the low-NaCl region (<400 mM), where reversible DNA unwrapping at both ends is expected. We observe differences, though, at higher NaCl concentration where the loss of FRET indicates permanent detachment of the DNA ends from the histone core, likely accompanied by H2A-H2B dimer eviction. The α -side was less stably bound ($c_{1/2}$ (E_{alpha}Dy_{beta}) = 631 ± 6 mM) than the β -side ($c_{1/2}$ (E_{beta}Dy_{alpha}) = 694 ± 12 mM). Thus, the $c_{1/2}$ values

indicate that hexasome formation proceeds preferentially by dimer eviction from the α -side. “

To make our conclusion more comprehensible, we added a new figure (Fig. 1) representing dye label positions and corresponding names.

Name (localization D, A)	H2B-Dy α	E α Dy β	E β Dy α	I β I α -wt	I β I α -mut
Traceable distance changes	Heterodimer and DNA position close to dyad axis	End of one DNA strand and DNA position close to dyad axis	End of the opposite DNA strand and DNA position close to dyad axis	Two internal positions on the DNA in wt nucleosomes	Two internal positions on the DNA in mutated nucleosomes
Purpose	Dimer release	Asymmetric DNA opening	Asymmetric DNA opening	Analysis of transient intermediates	Analysis of transient intermediates

Additionally, a detailed specification of our constructs can be found on pages 15-16 of the revised manuscript :

- E β Dy α DNA labeled at 5' end (E β) with A488 and at position -15 (close to the dyad) on the other strand (Dy α) with A594, plus wt octamers
- E α Dy β analogous to nucleosome E β Dy α with label positions on the complementary strands (A488 at 5' end α , A594 at +15).

Reviewer #1 could have been confused by another statement and $c_{1/2}$ values on page 9 (of the revised manuscript) concerning the difference in stability between mutated and non-mutated nucleosomes (I β I α -wt and I β I α -mut samples):

“Salt-induced nucleosome disassembly in ensemble FRET (Fig. 6B) confirmed that mutated nucleosomes were significantly less stable than wt nucleosomes: $c_{1/2}(I\beta I\alpha\text{-wt}) = 783 \pm 5$ mM, $c_{1/2}(I\beta I\alpha\text{-mut}) = 618 \pm 7$ mM (standard errors, N=6).”

We hope that the already mentioned new Figure 1 may help to avoid of such confusion.

3. When comparing their work with Ngo et al, they used the term 'perturbed' vs 'without external perturbation'. I know what they mean, i.e. with or without external tension applied. But in the current study, they use high salt concentration solution which might

also be considered perturbation. I would suggest they change it to external tension. It is actually quite likely that tension applied puts the nucleosome in the more physiologically relevant state compared to high salt condition.

Thank you for this remark; we admit that this term could be a bit confusing. Thus, we replaced “without external perturbation” or “unperturbed” by “without external tension” throughout the whole text.

Reviewer #2 (Remarks to the Author):

The manuscript by Gansen, et al. reports multiparameter single molecule fluorescence studies of Cy3-Cy5 labeled nucleosomes. They results on hexasome dynamics and the that nucleosome undergo rapid unwrapping with the H2A-H2B heterodimer attached to the DNA. The multiparameter single molecule fluorescence studies are ideal for studying nucleosome dynamics. So, I was looking forward to reading about this work on using their multiparameter fluorescence detection (MFD). However, this manuscript is (at least for me) extremely difficult to read. It is full of abbreviations that are not defined and it assumes the reader is already familiar with the experimental design. While this reviewer is familiar with the authors previous work on MFD and nucleosome studies, I was continually trying to find definitions of all of their abbreviations and nomenclature. It was taking so long to review the manuscript that I decided to stop and recommend that the authors rewrite the paper so it is readable and then resubmit the paper.

We thank for the reviewer's interest and apologize for the difficulty in reading. We understood the concerns of Reviewer 2 and tried to increase the readability of the manuscript by reduction the amount of used abbreviations and simplification of the used abbreviations and nomenclature. Hence, we rewrote and simplified, where possible, parts of the main manuscript to make it easier to follow and more accessible for a broader audience:

- 1) We converted the different donor-acceptor distances obtained by the different approaches and are now presenting only one type of distance: mean interdye distance obtained from intensity based FRET efficiency $\langle R_{DA} \rangle_E$ instead of earlier used lifetime based distances e.g. $\langle R_{D\alpha A} \rangle$ or intensity based ones as well as distances obtained by accessible volume (AV) simulations $R_{DA,mp}$ or R_{DA}^α .

“As the observed donor-acceptor (DA) distances are usually spatially averaged, the interdye distances recovered by fluorescence lifetime- or intensity-based FRET measurements differ slightly due to inherent different weighting. However, the knowledge of the AVs allows us to interconvert them (Materials and Methods and Supplementary Table 1). For convenience we present only the FRET-averaged DA distances, $\langle R_{DA} \rangle_E$.”

2) We simplified abbreviations:

- a. average fluorescence-weighted donor lifetime from $(\langle\tau_{D(A)}\rangle_F)$ to $(\langle\tau_D\rangle_F)$
- b. we changed the name of the dynamic FRET population MF' to MF* to make it easily distinguishable from the static population MF.
- c. we elaborated one of our major findings to make our conclusions more comprehensible (page 6 – in the revised manuscript):

“We have several corroborating arguments for the coexistence of MF and MF*: (i) Within the precision of FRET the 2D MFD data of donor lifetime and intensity-based FRET efficiency indicate that the inter-dye distance of MF* is slightly shorter (by $\approx 4 \text{ \AA}$) compared to MF (Fig.4B). Because the distances are close to R_0 , this corresponds to a significant change of FRET efficiency $\Delta E \approx 0.1$ that is readily detectable by sub-ensemble fluorescence decay analysis (Supplementary Figs. 1C-E). (ii) The dependence of the kinetic exchange rates on the overall nucleosome concentration, as described next, can be only explained by at least two species with very similar FRET properties: a dynamic (MF*) and a static (MF) subpopulation. (iii) Considering the disassembly pathway of mononucleosomes, mechanistic arguments postulate the existence of two structural states, octasome and hexasome, that have similar FRET efficiencies for the $I_{\beta}I_{\alpha}$ FRET-pair (see discussion).”

As already mentioned, we added an additional Figure1 illustrating the different dye labeling positions and the corresponding names, which should make our work more comprehensible. We hope that this helps to identify our constructs and dyes (Alexa488 as a donor and Alexa 594 and Cy5 as acceptors, respectively). - We deliberately did not use Cy3 for any of our experiments, because it very sensitive to its environment (PIFE effect), so that FRET experiments are very difficult to be interpreted quantitatively as needed in this work.

Furthermore, we optimized Figure 5 by fitting a model function to the data and added Fig. 5C to illustrate the salt dependence of the species fractions and rates.

During revision we discovered that the data originally shown in Fig. 6B was not properly normalized. This was corrected and the data was refitted, resulting in slightly but not significantly changed $c_{1/2}$ -values: $c_{1/2}(I_{\beta}I_{\alpha}\text{-wt}) = 783 \pm 5 \text{ mM}$ and $c_{1/2}(I_{\beta}I_{\alpha}\text{-mut}) = 618 \pm 7 \text{ mM}$ instead of $c_{1/2}(I_{\beta}I_{\alpha}\text{-wt}) = 800 \pm 29 \text{ mM}$ and $c_{1/2}(I_{\beta}I_{\alpha}\text{-mut}) = 633 \pm 23 \text{ mM}$, respectively.

To make our quantitative findings more accessible to the reader, we revised Figure 7 to improve the clarity of our findings. We added also Figure 8B where all equilibria and corresponding rate constants are listed which are presented in Figure 8A. To my best knowledge, these most numbers are new. When it was

possible we compared them to literature to demonstrate their consistency (e.g. for step I (e.g. work by Lee et al in Ref. 18,40)

Moreover, we tried to structure results and discussion sections together with short summaries of our findings to demonstrate their overall consistency and clarify their meaning in the context of nucleosome disassembly.

REVIEWERS' COMMENTS:

Reviewer #3 (Remarks to the Author):

Review of "Reversible transitions in nucleosomes on the microsecond to minute time scale revealed by high precision single-molecule FRET" by Alexander Gansen, Suren Felekyan, Ralf Kuhnemuth, Katalin Toth, Claus A.M. Seidel, Jorg Langowski.

Review. What is probably the last paper from a strong group. The paper has solid data but is suffering from the lack a clear description of the problem. Salt titrations have a long history in chromatin biology and understanding the formation of nucleosomes under these conditions is important but you have to be clear about differences in this how nucleosomes might be formed in cells. Therefore it needs to be clear that the equilibrium inferred from changes in populations as a function of salt is not the same as a true equilibrium. Or you need to account for ions in your model. Other than this it is a solid and interesting paper.

Major comments:

"Single-molecule (sm) techniques allow one to perform a detailed structural and kinetic analysis of transient structures in an ensemble under equilibrium conditions" They need to show this or cite it. Because nucleosomes alone are kinetically controlled. Again "under equilibrium conditions on hitherto unprecedented short time scales in the microsecond range. " there is no evidence of equilibrium outside of 'substate' or partial disassociation. If they are going to say this they need to show that it can reform. Which they currently don't. Unless you count changing the salt concentration equilibrium; which you shouldn't.

Minor

Reword "simply be due to the non-equilibrium conditions inherent to stopped flow experiments as performed in 30. "

NCOMMS-17-05740A

Answers to the REVIEWERS' COMMENTS:

The authors thank for the interest in our work as well as for the prompt and profound revision of this manuscript. Please see below our answers to the comments.

Reviewer #3 (Remarks to the Author):

Review of "Reversible transitions in nucleosomes on the microsecond to minute time scale revealed by high precision single-molecule FRET" by Alexander Gansen, Suren Felekyan, Ralf Kuhnemuth, Kathrin Lehmann, Katalin Toth, Claus A.M. Seidel, Jorg Langowski.

Review. What is probably the last paper from a strong group. The paper has solid data but is suffering from the lack a clear description of the problem. Salt titrations have a long history in chromatin biology and understanding the formation of nucleosomes under these conditions is important but you have to be clear about differences in this how nucleosomes might be formed in cells. Therefore it needs to be clear that the equilibrium inferred from changes in populations as a function of salt is not the same as a true equilibrium. Or you need to account for ions in your model. Other than this it is a solid and interesting paper.

Answer:

In order to clarify the relevance of the results obtainable in conditions far from physiological, we moved a corresponding sentence from the discussion into the introduction.

"One could argue that the conditions, up to > 1 M NaCl, are often outside the physiological range, but the detected intermediates may be also relevant for in vivo dissociation pathways."

Major comments:

"Single-molecule (sm) techniques allow one to perform a detailed structural and kinetic analysis of transient structures in an ensemble under equilibrium conditions" They need to show this or cite it. Because nucleosomes alone are kinetically controlled. Again "under equilibrium conditions on hitherto unprecedented short time scales in the microsecond range. "there is no evidence of equilibrium outside of 'substate' or partial disassociation. If they are going to say this they need to show that it can reform. Which they currently don't. Unless you count changing the salt concentration equilibrium; which you shouldn't.

Answer:

If the sample in stable solution is stable and the measurement time is long enough, it has shown be many groups that single-molecule techniques are perfectly suited to study the kinetics of exchange reactions in biomolecular systems (e.g. the recent review "Toward dynamic structural biology: Two decades of single-molecule Förster resonance energy transfer" Lerner et al. Science 359, eaan1133, **2018**).

Regarding the equilibrium: At NaCl concentrations smaller than ~ 800 mM (see Supplementary Fig. 3), the sample is sufficiently stable. Under these conditions, we can measure steady state concentrations of nucleosome species (octasome, hexasome and tetrasome) that depend on total nucleosome concentration. Thus, these species are in equilibrium as shown by our time dependent measurements (e.g. Fig. 6c, d and Supplementary Figs. 4 d-g).

However, at higher NaCl concentrations we did not directly measure reformation of nucleosomes from completely dissociated subunits (DNA and histones). That is a very

slow process (as indicated by the measured small off rates (Supplementary Figure 4h, i) and was not addressed in our study. To conclude, only for completely dissociated species we cannot prove an equilibrium.

We added the following sentences for clarification:

On page 2:

"If the sample is stable solution is stable and the measurement time is long enough, it has shown be many groups that single-molecule techniques are perfectly suited to study the kinetics of exchange reactions in biomolecular systems ³³".

On page 11:

"Note that at NaCl concentrations > 800 mM tetrasomes are formed more easily (Supplementary Fig. 3). Tetrasome formation and subsequent tetrasome dissociation are both irreversible under single-molecule conditions (nM concentrations), in contrast to the stable equilibrium conditions at lower NaCl concentrations".

Minor

Reword "simply be due to the non-equilibrium conditions inherent to stopped flow experiments as performed in 30. "

Answer:

The sentence has been reworded:

"This could imply different opening mechanisms in high and low salt or simply caused by the non-equilibrium conditions inherent to stopped flow experiments reported in reference ³⁰"